# Proteomic and Biochemical Analyses of the Mechanism of Tolerance in Mutant Soybean Responding to Flooding Stress

**DOI:** 10.3390/ijms22169046

**Published:** 2021-08-22

**Authors:** Setsuko Komatsu, Hisateru Yamaguchi, Keisuke Hitachi, Kunihiro Tsuchida, Yuhi Kono, Minoru Nishimura

**Affiliations:** 1Faculty of Environment and Information Sciences, Fukui University of Technology, Fukui 910-8505, Japan; 2Department of Medical Technology, Yokkaichi Nursing and Medical Care University, Yokkaichi 512-8045, Japan; h-yamaguchi@y-nm.ac.jp; 3Institute for Comprehensive Medical Science, Fujita Health University, Toyoake 470-1192, Japan; hkeisuke@fujita-hu.ac.jp (K.H.); tsuchida@fujita-hu.ac.jp (K.T.); 4Central Region Agricultural Research Center, National Agriculture and Food Research Organization, Joetsu 943-0193, Japan; k41523@affrc.go.jp; 5Graduate School of Science and Technology, Niigata University, Niigata 950-2181, Japan; nisimura@agr.niigata-u.ac.jp

**Keywords:** proteomics, mutant soybean, flooding, glycoprotein folding, fermentation, cell death

## Abstract

To investigate the mechanism of flooding tolerance of soybean, flooding-tolerant mutants derived from gamma-ray irradiated soybean were crossed with parent cultivar Enrei for removal of other factors besides the genes related to flooding tolerance in primary generated mutant soybean. Although the growth of the wild type was significantly suppressed by flooding compared with the non-flooding condition, that of the mutant lines was better than that of the wild type even if it was treated with flooding. A two-day-old mutant line was subjected to flooding for 2 days and proteins were analyzed using a gel-free/label-free proteomic technique. Oppositely changed proteins in abundance between the wild type and mutant line under flooding stress were associated in endoplasmic reticulum according to gene-ontology categorization. Immunoblot analysis confirmed that calnexin accumulation increased in both the wild type and mutant line; however, calreticulin accumulated in only the mutant line under flooding stress. Furthermore, although glycoproteins in the wild type decreased by flooding compared with the non-flooding condition, those in the mutant line increased even if it was under flooding stress. Alcohol dehydrogenase accumulated in the wild type and mutant line; however, this enzyme activity significantly increased and mildly increased in the wild type and mutant line, respectively, under flooding stress compared with the non-flooding condition. Cell death increased and decreased in the wild type and mutant line, respectively, by flooding stress. These results suggest that the regulation of cell death through the fermentation system and glycoprotein folding might be an important factor for the acquisition of flooding tolerance in mutant soybean.

## 1. Introduction

Along with climate changes, the increased tendency of flooding has triggered severe crop reductions in yield and quality around the world [1]. Flooding is classified into two forms, which are waterlogging and submergence, depending on water depth [2]. Waterlogging is the condition that water exists on the soil surface and only plant roots are surrounded by water, while submergence is the state that the whole plant partially or completely immerses in water [3]. Flooding is a major threat causing substantial yield decline of crops and is expected to be even more serious in many parts of the world due to climatic anomalies in the future. In Japan, more than 80% of soybeans are cultivated in fields converted from paddy fields. During the rainy season, from June to the middle of July, soybean seedlings are damaged due to submergence in fields with poor drainage. Understanding the mechanisms of plants coping with unanticipated flooding is crucial for developing new flooding-tolerant crop varieties [4]. These previous findings indicated that the development of flooding-tolerant crop is an important task for improvement of crop yield.

In rice, which is one of the few flooding-tolerant crops, seedlings cannot grow under extended periods of complete submergence [5]. One of the two adaptive strategies for flooding tolerance is the *SNORKEL 1**/2* dependent escape strategy [6]. It promotes the internode elongation through the stimulation of gibberellin biosynthesis in deep-water rice under flooding stress, and thereby enabling rice grows upward to the water surface for air exchange [6]. Another adaptive strategy is the *S**UBMERGENCE1* dependent quiescence strategy [7]. In a few rice-tolerant varieties against submergence, the elongation of stem and leaf is inhibited by suppressing the increase in ethylene concentration, which decreases the sensitivity of plants to gibberellin [7]. Although the mechanisms of flooding tolerance in rice are well reported, those of soybean are still unclear, because the flooding-tolerant materials such as mutant lines or cultivars are not enough.

Although soybean is one of the major agricultural crops, it is particularly sensitive to flooding stress [8]. Plant growth and grain yield of soybean are markedly reduced in flooded soil [8]. When soybean was exposed to flooding at the vegetative growth stage or reproductive growth stage, grain yield and quality were reduced [9]. In addition, secondary aerenchyma was formed and worked as an oxygen pathway under flooded conditions [10]. Furthermore, flooding stress impaired plant growth by inhibiting root elongation and reducing hypocotyl pigmentation [11]. Under flooding, soybean seedlings showed differential regulation of proteins involved in signal transduction, hormonal signaling, transcriptional control, glucose degradation/sucrose accumulation, alcohol fermentation, gamma-aminobutyric acid shunt, suppression of reactive-oxygen species scavenging, mitochondrial impairment, ubiquitin/proteasome-mediated proteolysis, and cell-wall loosening [12,13,14,15]. Although flooding-response mechanisms in soybean were reported, characterization of the mechanism of flooding tolerance is needed regarding to agricultural usage.

To characterize the mechanism of flooding tolerance in soybean, flooding-tolerant lines were generated by physical mutagenesis of gamma-ray irradiation [16]. For preparation of the flooding-tolerant mutant [16], flooding-tolerant tests were repeated five times using gamma-ray irradiated soybeans, whose root growth (M6 stage) was not suppressed even if it was under flooding stress. Using the primary generated flooding-tolerant mutant, gel-based proteomic analysis was performed, indicating that activation of the fermentation system in the early stages of flooding is an important factor for the acquisition of flooding tolerance in soybean [16]. Using this mutant soybean and abscisic-acid treated soybean, which show flooding tolerance, proteomics [17], transcriptomics [18], and metabolomics [19] were performed at the initial stage of flooding stress. In this study, to remove other factors besides the genes related to flooding tolerance in primary generated mutant soybean, it was crossed with parent cultivar Enrei and flooding-tolerant tests were repeated two times. Using the progeny of this cross, morphological analysis was performed under flooding stress. Based on the morphological results, gel-free/label-free proteomic analysis was carried out to explore the mechanism of tolerance for the positive effects on growth of mutant soybean treated with flooding. Proteomic results were subsequently confirmed by immunoblot, enzyme activity, and physiological analyses.

## 2. Results

### 2.1. Morphological Analysis of Mutant Soybean under Flooding Stress

To remove other factors besides the genes related to flooding tolerance in primary generated mutant soybean [16], they were crossed with parent cultivar Enrei. Mutant lines with flooding tolerance selected from progeny were used in this study. As mutant lines, 1386-6 (G2), 1386-9 (G3), 1387-12 (G4) lines were used for morphological analysis. To investigate the morphology of mutant soybean under flooding stress, morphological changes of soybean were analyzed (Figure 1). To induce flooding stress, water was added to immerse 2-day-old soybeans, and samples were collected at 6 days after 5-days flooding (Figure 1). The length/weight of stem/root of the mutant lines did not change compared with the wild type without flooding stress (Figure 2). The length/weight of stem/root of the wild type soybean was significantly suppressed by flooding; however, those of the 1386-6 (G2) and 1386-9 (G3) lines were better than those of the wild type even if they were treated with flooding. By flooding, length and weight of hypocotyl did not change between wild type and mutants; however, those of epicotyls increased in the mutant compared with wild type (Figure 2). Based on morphological results, 1386-6 (G2) line of mutant was used for proteomic analysis.

### 2.2. Identification and Functional Investigation of Proteins in Mutant Soybean under Flooding Stress

In order to explore the cellular mechanism on growth of mutant soybean, a gel-free/label-free proteomic technique was used. Proteins were extracted from the root including hypocotyl after 2-days flooding of 2-day-old soybeans (Figure 1). Totally, 7889 proteins were detected by MS analysis. Four kinds of group, which were wild type/mutant line under flooding/non-flooding, were prepared. The criteria for significantly changed proteins were 2 or more than 2 matched peptides with a *p*-value less than 0.05 (Appendix A). Relative abundance of 6967 proteins from wild type under flooding was compared with that from non-flooding. Among the differentially changed 986 proteins, 514 proteins increased and 472 proteins decreased in wild type under flooding stress compared with under non-flooding (Appendix A). Furthermore, relative abundance of 6967 proteins from mutant under flooding was compared with that from non-flooding. Among the differentially changed 833 proteins, 350 proteins increased and 483 proteins decreased in the mutant line under flooding stress compared with non-flooding (Appendix A). The proteomic data of all samples from different groups were compared by PCA, which indicated the different accumulation patterns of proteins from different treatment (Appendix A). Flooding stress largely affected soybean proteins in both wild type and mutant line (Figure 3).

Furthermore, the abundance of proteins differentially changed with fold change ≥1.5 and ≤1/3 in wild type or mutant line under flooding stress compared with non-flooding were categorized using cellular component of gene-ontology analysis (Figure 3). Among the differentially changed 573 proteins, 294 proteins increased and 279 proteins decreased in wild type under flooding stress compared with non-flooding (Figure 3 left). Among the differentially changed 454 proteins, 169 proteins increased and 285 proteins decreased in the mutant line under flooding stress compared with non-flooding (Figure 3 right). Differentially increased proteins were mainly located in cytoplasm and decreased proteins were in membrane. Endoplasmic reticulum related proteins were oppositely changed between the wild type and mutant line under flooding condition (Figure 3). Based on proteomic results, proteins in oppositely changed subcellular category, which is endoplasmic reticulum, were confirmed by biochemical and physiological analyses.

### 2.3. Immunoblot Analysis of Proteins Involved in Endoplasmic Reticulum in Mutant Soybean under Flooding Stress

To better uncover the change of proteins from different treatments, immunoblot analysis of proteins in endoplasmic reticulum was performed (Figure 4). Proteins were extracted from root including hypocotyl of soybeans, which were wild type and mutant line treated with or without flooding. Coomassie brilliant blue staining pattern was used as loading control (Appendix A). To investigate the change of proteins in endoplasmic reticulum, the accumulation of calnexin and calreticulin was analyzed (Appendix A). Immunoblot analysis confirmed that calnexin accumulation increased in both wild type and mutant line; however, calreticulin accumulated in only mutant line under flooding stress (Figure 4A,B). Furthermore, Concanavalin A antibody was used for lectin blot (Appendix A); and the accumulation of glycoproteins decreased in wild type by flooding and that in the mutant line under flooding stress even increased in the level of the non-flooding (Figure 4C,D). In N-linked glycoproteins, exostosin domain-containing protein (I1JXR7) was significantly decreased in wild type by flooding stress (Appendix A); however, it was increased in the mutant line even if it was under flooding stress (Appendix A). These results indicated that folding of glycoproteins was improved in the mutant line, even if it was flooding condition.

### 2.4. Alcohol Dehydrogenase (ADH) Accumulation and Activity in Mutant Soybean under Flooding Stress

To better uncover the change of fermentation in different treatments, change of ADH was analyzed. Proteins were extracted from root including hypocotyl of soybeans, which were wild type and mutant line treated with or without flooding. Coomassie brilliant blue staining pattern was used as loading control (Appendix A), and the accumulation of ADH was analyzed with immunoblot (Appendix A). ADH accumulated in wild type and mutant line by flooding stress (Figure 5A). Furthermore, in order to understand the molecular mechanism of the mutant line under flooding stress, ADH activity was analyzed. Enzyme activity of ADH significantly increased and mildly increased in wild type and mutant line, respectively, under flooding stress (Figure 5B). These results indicated that the fermentation was caused in flooded seedlings and it was recovered in the mutant line, even if it was flooding condition.

### 2.5. Evans-Blue Staining with Mutant Soybean under Flooding Stress

To better understand suppression of root growth, the cell death of root-tip was evaluated by Evans-blue staining (Figure 6). Seedlings after 2 days of flooding treatment were stained with Evans blue, and the Evans blue was extracted into the solvent from the stained root tips and measured. Evans-blue uptake in root tip increased in wild type by flooding stress and decreased in the mutant line (Figure 6). These results indicated that the suppression of root growth by flooding was caused by cell death of root tip induced by flooding stress in wild type and it has not happened in the mutant line.

## 3. Discussion

### 3.1. Mutation in Flooding-Tolerant Soybean Has Positive Effects on the Growth of Plant under Flooding Stress

To investigate the difference of mutant soybean before and after being crossed with parent cultivar Enrei, morphological changes of the mutant lines were analyzed (Figure 2) and compared with previous results [16]. Before being crossed with cultivar Enrei, these mutant soybean lines exhibited restricted growth when exposed to flooding conditions for 7 days, whereas no wild-type plants survived this condition [16]. When the water was removed, however, the mutant lines started to grow again [16]. In this study, although the growth of the wild type was significantly suppressed by flooding compared with non-flooding condition, that of mutant lines was better than that of the wild type even if it was treated with flooding (Figure 2). Taken together, this study in conjunction with the previous report [16] indicates that mutant lines keep the morphological improvement after being crossed with parent cultivar Enrei. 

Bottlenecks in genetic diversity challenge germplasm screening for flood-tolerant cultivars of soybean, while proteomics and metabolomics assist in the elucidation of flooding tolerance in soybean. Nanjo et al. [20] reported that RNA binding/processing-related proteins and flooding stress indicator proteins correlated with flood-tolerant index. In addition, Li et al. [21] reported that secondary metabolites such as isoflavonoid improved flooding tolerance cultivar. However, these factors did not exist in flooding-tolerant mutant in this study. In the case of rice, there are two types showing tolerance against flooding stress, which are an escape strategy [6,22] and a quiescence strategy [7,23]. Because *SNORKEL 1/2* and *SUBMERGENCE 1* genes in rice do not exist in soybean, the mechanism of flooding tolerance might be completely different in soybean.

### 3.2. Regulation of Alcohol-Formation Pathway Is Related to the Mechanism of Flooding Tolerance

In the flooding environment, the energy to maintain plant vitality mainly relies on the ethanol metabolic pathway in glycolysis to degrade glucose and glycogen accompanied by ATP generation. ADH and pyruvate decarboxylase are key to the establishment of the fermentative metabolism in plants during oxygen shortage [24,25]. The flooding tolerance of plants was proportional to the change in ADH activity in response to flooding. Due to the oxygen-depleted environment arising from flooding, a crisis in ATP availability occurred; however, to manage the energy crisis, the balance of glycolysis aided in soybean survival from submergence [17,21] and enhanced fermentation was necessary for the acquisition of flooding tolerance [16]. These findings indicate that activated fermentation and glycolysis confer soybean-flooding tolerance to ensure survival.

The respective overexpression of *ThADH1* and *ThADH4* in *Populus* [26] and/or *GmAdh2* in soybean [27] improved the growth of transgenic soybeans under flooding stress. *ADH* genes were regarded as important candidates for genetic manipulation to achieve flooding tolerance in plants, through improving the adaptability to hypoxia [28]. In this study, 3 proteins having ADH activities were identified (Appendix A), which were “uncharacterized protein” by search with UniProtKB Glycine max (SwissProt TreEMBL). ADH accumulation increased in both wild type and mutant line under flooding stress; however, ADH activity significantly increased and mildly increased in wild type and mutant line, respectively, under flooding stress (Figure 5). These results suggest that mutant line can survive from flooding through the mild activation of the alcohol-fermentation pathway.

### 3.3. Glycoprotein Folding Is Related to the Mechanism of Flooding Tolerance

Based on proteomic results, oppositely changed proteins determined in the endoplasmic reticulum (Figure 3) were confirmed by immunoblot analysis (Figure 4). Endoplasmic reticulum mediates protein folding and assembly through a well-coordinated system of chaperones such as calnexin, protein disulfide isomerase, and heat shock proteins [29]. Among these proteins, calnexin is involved in protein folding and quality control [30]. Calreticulin, which is a major calcium binding chaperone in the endoplasmic reticulum, is a key component of the calnexin/calreticulin cycle [31]. Furthermore, two calreticulins existed in rice [32] and phosphorylated calreticulin showed the molecular weight of 56 kDa [28]. The calnexin/calreticulin cycle is responsible for the folding of newly synthesized proteins, especially glycoproteins, for quality control and stability in the endoplasmic reticulum [31]. In this study, heat shock 70 kDa proteins as well as calreticulin and calnexin were identified by proteomic analysis (Appendix A). The present results with previous reports indicate that the activation by the phosphorylation of both calreticulin and calnexin as well as heat shock protein 70 is needed for glycoprotein folding. This study suggests that glycoproteins might not be folded under flooding stress, because only calnexin increased in calnexin/calreticulin cycle.

In soybean, calnexin and calreticulin decreased, which led to the reduced accumulation of glycoproteins and disruption of endoplasmic reticulum homeostasis under flooding and drought stresses [33]. The accumulation of calnexin/calreticulin and glycoproteins significantly increased in soybean under flooding with silver nano particles with other chemicals, which improved soybean growth [34]. In this study, lectin legB domain-containing protein and glycosyltransferase were also found by proteomic analysis (Appendix A). Furthermore, in N-linked glycoproteins, exostosin domain-containing protein significantly decreased in wild type by flooding stress; however, it was recovered in the mutant line even if it was under flooding stress (Tables S1–3). Exostosin domain-containing protein is in Golgi apparatus and related to secondary cell-wall biogenesis. The root and hypocotyl of soybean caused the suppression of lignification through the decrease in glycoproteins by downregulation of reactive oxygen species and jasmonate biosynthesis under flooding stress [35]. Because the abundances of calnexin, calreticulin, and glycoproteins increased in mutant soybean under flooding (Figure 4), the folding of glycoproteins might be improved in the mutant line with the increase in both calnexin and calreticulin, even if it was flooding condition. These results with previous reports suggest that the folding of glycoproteins is essential for recovery from damage caused by flooding stress.

### 3.4. Cell Death in Soybean Root Is Related to Flooding Tolerance

At the initial stage of flooding, proteomic studies using flood-tolerant materials of mutant and ABA-treated soybeans showed that protein synthesis and RNA regulation-related proteins triggered soybean tolerance [17]. Together with RNA regulation and protein metabolism, hormone response contributed to initial flooding tolerance through the inhibition of cytochrome P450 77A1 [18]. Integrated omic data derived from proteomics and metabolomics indicated that fructose was a critical metabolite to assist in soybean flooding tolerance at initial stages via regulation of hexokinase and phosphofructokinase [19]. Taken together, proteomics, in combination with transcriptomics and metabolomics, improves the capability to uncover prospective flood tolerance responses in soybean as for RNA modification, protein synthesis, and sugar catabolism at the initial stage of flooding. However, 4-day-old mutant soybean with 2-days flooding stress [36], did not show same mechanisms compared with flooding stress at the initial stage.

The loss of root tips by flooded seedlings was caused by flooding-induced root tip cell death and it was recovered in the mutant line (Figure 6). Within cells, protein synthesis and degradation are well balanced, because a small decrease in synthesis or increase in degradation results in cell death [37]. The ubiquitin proteasome system responds to the stress conditions by regulating the degradation of misfolded proteins to restrict their accumulation [38]. Autophagic programmed cell death of the meristematic cells has been implicated in root-tip death of several species, including pea and maize exposed to severe stress conditions [39]. The death program was triggered by an imbalance between folded and unfolded proteins in the endoplasmic reticulum, which represents a common cellular stress [40]. In this study, glycoproteins did not fold under flooding stress; however, the folding of glycoproteins was improved in the mutant line with the increase in both calnexin and calreticulin (Figure 6). These results with previous reports suggest that increased accumulation of misfolded related proteins causes cell death, which might reduce the growth of soybean under flooding stress. On the other hand, mutation in flooding-tolerant soybean might improve the quality of proteins; as a result, cell death is suppressed.

## 4. Materials and Methods

### 4.1. Plant Material and Experimental Design 

Experimental design for investigation of the mechanism of flooding tolerance in mutant soybean was summarized in Figure 1. Flooding-tolerant founder mutant soybeans [16] were crossed with wild type soybean (*Glycine max* L. cultivar Enrei). Mutant lines with flooding-tolerant selected from progeny were used. As mutant lines, 1386-6 (G2), 1386-9 (G3), 1387-12 (G4) lines were used in this study.

Seeds were sterilized with 2% sodium hypochlorite solution, rinsed twice in water, and sown in 400 mL of silica sand in a seedling case. Soybeans were grown at 25 °C and 60% humidity under white fluorescent light (160 µmol m^−2^ s^−1^, 16 h light period/day). To induce flooding stress, water was added above the soil surface to immerse 2-day-old soybeans. For morphological analysis, root and hypocotyl were collected from at 6 days after 5-days flooding. For proteomic analysis and other confirmation experiments, roots including hypocotyl were collected from 2-days flooded soybeans. 

Three independent experiments were performed as biological replications for all experiments, meaning that the seeds were sown on different days. A total of 14 seeds were sown evenly in each seedling case. The statistical significance of multiple groups was evaluated by one-way ANOVA test. SPSS 20.0 (IBM, Chicago, IL, USA) statistical software was used for the evaluation of the results. A *p*-value of less than 0.05 was considered as statistically significant.

### 4.2. Protein Extraction, Protein Enrichment, Reduction, Alkylation, and Digestion

Protein extraction was performed with methods described in the previous study [41]. The method of Bradford [42] was used to determine the protein concentration with bovine serum albumin used as the standard. Extracted proteins (100 µg) were adjusted to a final volume of 100 µL. Protein enrichment, reduction, alkylation, and digestion were performed with methods described in the previous study [41]. 

### 4.3. Protein Identification Using Nano-Liquid Chromatography Mass Spectrometry 

The peptides were loaded onto the LC system (EASY-nLC 1000; Thermo Fisher Scientific, San Jose, CA, USA) equipped with a trap column (Acclaim PepMap 100 C18 LC column, 3 µm, 75 µm ID × 20 mm; Thermo Fisher Scientific). The liquid chromatography (LC) conditions as well as the mass spectrometry (MS) acquisition conditions were described in the previous study [43].

### 4.4. Mass Spectrometry Data Analysis

The MS/MS searches were carried out using MASCOT (version 2.6.1, Matrix Science, London, UK) and SEQUEST HT search algorithms against the UniprotKB *Glycine*
*max* (SwissProt TreEMBL, TaxID = 3847, version 2020-09-18) using Proteome Discoverer (PD) 2.2 (version 2.2.0.388; Thermo Fisher Scientific). The workflow for both algorithms included spectrum files RC, spectrum selector, MASCOT, SEQUEST HT search nodes, percolator, ptmRS, and minor feature detector nodes. Oxidation of methionine was set as a variable modification and carbamidomethylation of cysteine was set as a fixed modification. Mass tolerances in MS and MS/MS were set at 10 ppm and 0.6 Da, respectively. Trypsin was specified as protease and a maximum of 2 missed cleavage was allowed. Target-decoy database searches used for calculation of false discovery rate (FDR) and for peptide identification FDR was set at 1%.

### 4.5. Differential Analysis of Proteins Using Mass Spectrometry Data

Label-free quantification was also performed with PD 2.2 using precursor ions quantifiler nodes. For differential analysis of the relative abundance of peptides and proteins between samples, the free software PERSEUS (version 1.6.14.0, Max Planck Institute of biochemistry, Martinsried, Germany) [44] was used. The condition of analysis is described in the previous study [43]. Principal component analysis (PCA) was performed with PD 2.2.

### 4.6. Bioinformatic Analyses of Protein Functional Categorization

The sequences of the differentially accumulated proteins were subjected to a BLAST query against the gene-ontology database (http://www.geneontology.org/ (accessed on 19 September 2019)). 

### 4.7. Immunoblot Analysis

SDS-sample buffer consisting of 60 mM Tris-HCl (pH 6.8), 2% SDS, 10% glycerol, and 5% dithiothreitol was added to protein extracts. As protein marker, Precision Plus Protein Standard (Bio-Rad, Hercules, CA, USA) was used. Quantified proteins (10 µg) were separated by electrophoresis on a 10% SDS-polyacrylamide gel and transferred onto a polyvinylidene difluoride membrane using a semidry transfer blotter (Nippon Eido, Tokyo, Japan). The blotted membrane was blocked for 5 min in Bullet Blocking One reagent (Nacalai Tesque, Kyoto, Japan). After blocking, the membrane was cross-reacted with a 1: 1000 dilution of the primary antibodies for 1 h at room temperature. As primary antibodies, anti- ADH [45], calnexin [46], and calreticulin [47] antibodies were used. Anti-rabbit IgG conjugated with horseradish peroxidase (Bio-Rad) was used as the secondary antibody. For lectin blot, peroxidase-Concanavalin A antibody (Seikagaku, Tokyo, Japan) was used. After 1 h incubation, signals were detected using TMB Membrane Peroxidase Substrate Kit (Seracare, Milford, MA, USA) following protocol from the manufacturer. Coomassie brilliant blue staining was used as loading control. Raw data were saved TIFF format (16-bit gray scale image) and the integrated densities of bands were calculated using Image J software (version 1.8; National Institutes of Health, Bethesda, MD, USA). 

### 4.8. ADH Activity Assay

ADH activity was analyzed using ADH Activity Colorimetric Assay Kit (BioVision, Milpitas, CA, USA). A portion (50 mg) of samples was homogenized in 200 μL of ice-cold assay buffer and centrifuged at 13,000× *g* for 10 min at 4 °C to remove insoluble material. Extracts (50 μL) were added in 100 μL of reaction mixture containing 82 μL of ADH assay buffer, 8 μL of developer, and 10 μL of substrate. Mixture was incubated for 2 and 10 min at 37 °C and the absorbance of mixture was measured at 450 nm.

### 4.9. Evaluation of Cell Death Using Evans-Blue Dye

Cell death was evaluated by Evans-blue staining as described by Jacyn-Baker and Mock [48]. 

## 5. Conclusions

To remove other factors besides the genes related to flooding tolerance in previous isolated from gamma-ray irradiated soybean [16], mutants were crossed with the parent cultivar Enrei. To investigate the mechanism of flooding tolerance of soybean, flooding-tolerant mutants generated in this study were analyzed using a gel-free/label-free proteomic technique. Furthermore, proteomic results were confirmed with biochemical techniques. The main findings are as follows: (i) the growth of the mutant line was better than that of the wild type even if it was treated with flooding; (ii) oppositely changed proteins between the wild type and mutant lines under flooding stress were associated in endoplasmic reticulum; (iii) calnexin accumulation increased in the wild type and mutant line but calreticulin accumulated in only the mutant line under flooding stress; (iv) glycoproteins in the mutant line increased compared with those of the wild type even if it was under flooding stress; (v) ADH accumulated in the wild type and mutant line but enzyme activity significantly increased and mildly increased in the wild type and mutant line, respectively, under flooding stress; (vi) cell death increased and decreased in the wild type and mutant line, respectively, by flooding stress. These results suggest that regulation of cell death through the fermentation system and glycoprotein folding under flooding might be an important factor for the acquisition of flooding tolerance in soybean.

## Figures and Tables

**Figure 1 ijms-22-09046-f001:**
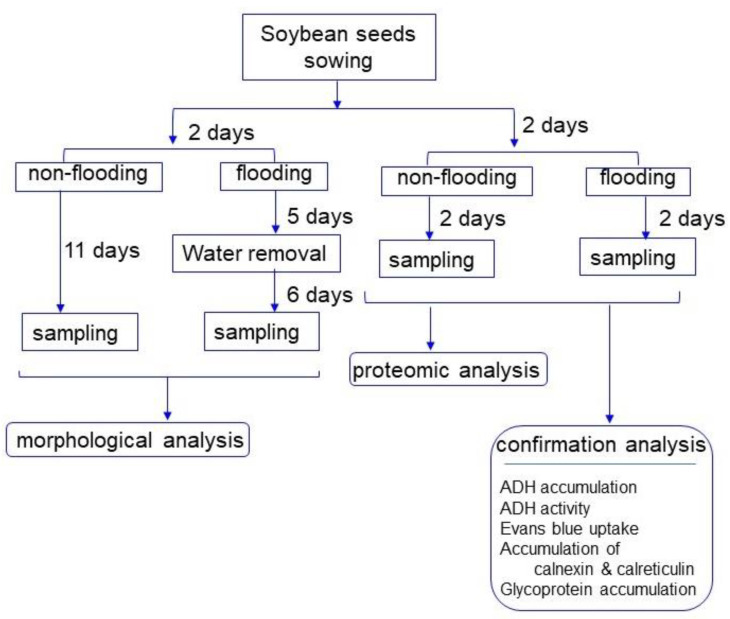
Experimental design for investigation of the mechanism of flooding tolerance in mutant soybean. As mutant lines, 1386-6 (G2), 1386-9 (G3), and 1387-12 (G4) lines were used. Soybean seedlings were analyzed with morphological and proteomics. Proteomic results were subsequently confirmed by immunoblot, enzyme activity, and physiological analyses. All experiments were performed with 3 independent biological replicates.

**Figure 2 ijms-22-09046-f002:**
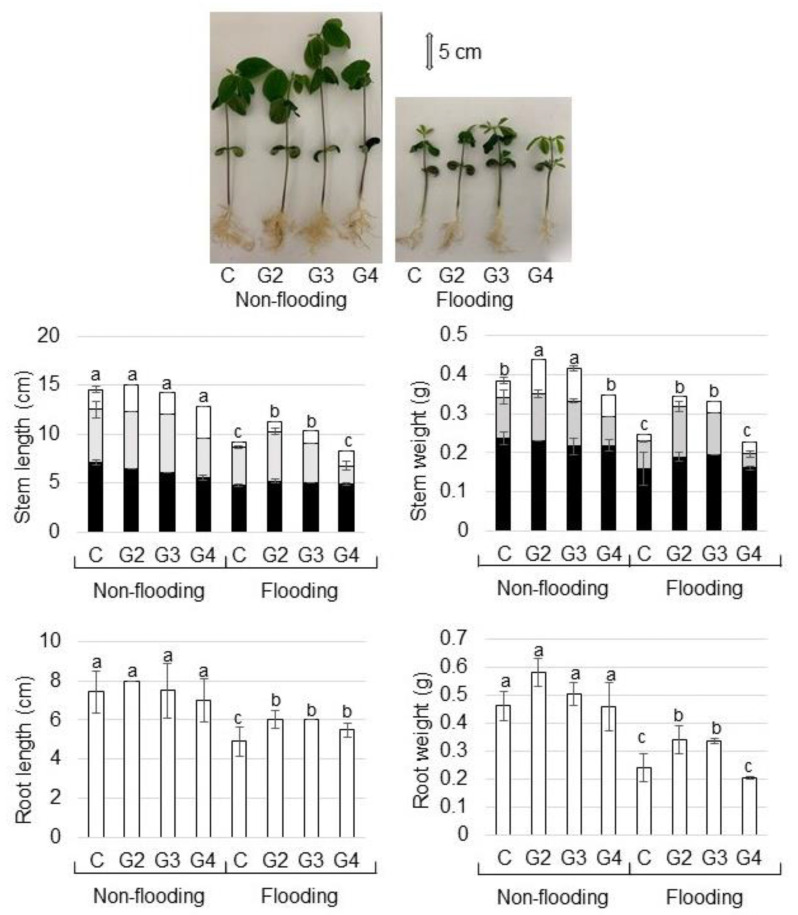
Morphological effect of flooding stress in mutant soybean. Wild type (C) and mutant lines, which are 1386-6 (G2), 1386-9 (G3), 1387-12 (G4), were used. Two-day-old soybeans were flooded for 5 days and water was removed. After 6 days of water removal, root and hypocotyl were collected. For non-flooded group, samples were collected at 13 days after sowing. Photograph shows 13-day-old soybeans. Hypocotyl (black column)/epicotyl (gray column)/2nd internode (white column) length, main root length, hypocotyl/epicotyl/2nd internode weight, and total root weight were measured as morphological parameters. In the graph, hypocotyl/epicotyl/2nd internode was summarized as stem. Data are shown as means ± SD from 3 independent biological replicates. The different letters indicate significant changes according to one-way ANOVA followed by Tukey’s multiple comparisons (*p* < 0.05).

**Figure 3 ijms-22-09046-f003:**
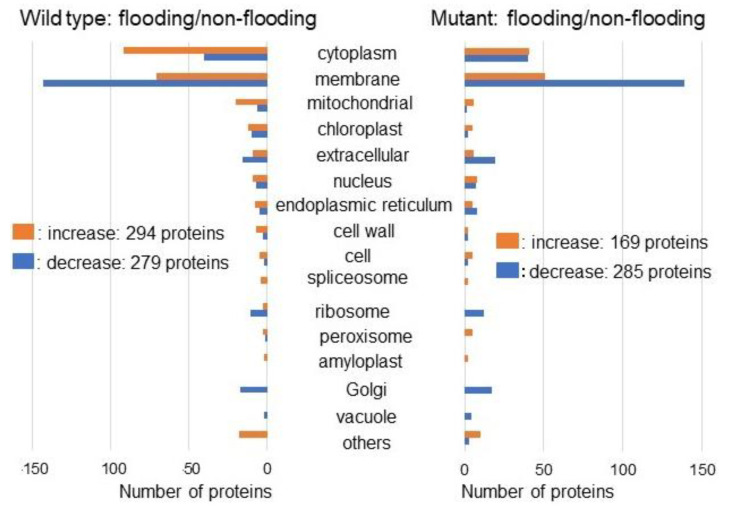
Gene-ontology categories of proteins with differential abundance in the mutant line and wild type treated with flooding. Two-day-old soybeans of the mutant line and wild type were exposed without (non-flooded) or with (flooded) flooding stress. After proteomic analysis, significantly changed proteins (*p* < 0.05) in the mutant line and wild type were compared between flooding and non-flooding conditions. Functional categories of changed proteins were determined using gene-ontology analysis (Appendix A). Red and blue columns show increased and decreased proteins, respectively.

**Figure 4 ijms-22-09046-f004:**
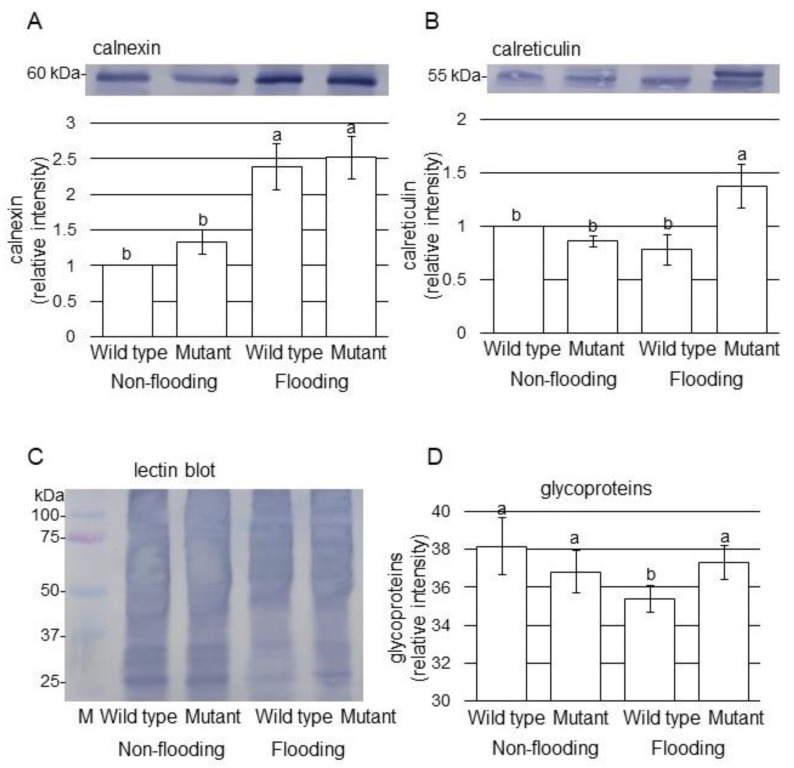
Immunoblot analysis of proteins involved in endoplasmic reticulum. Proteins were extracted from root including hypocotyl, separated on SDS-polyacrylamide gel by electrophoresis, and transferred onto membrane. The membrane was cross-reacted with anti-calnexin (**A**) and calreticulin (**B**) antibodies. In the case of glycoproteins, lectin-blot (**C**,**D**) was performed. Coomassie brilliant blue staining pattern was used as loading control (Appendix A). The integrated densities of bands were calculated using ImageJ software. Data are shown as the means ± SD from 3 independent biological replicates (Appendix A). The different letters indicate significant changes according to one-way ANOVA followed by Tukey’s multiple comparisons (*p* < 0.05).

**Figure 5 ijms-22-09046-f005:**
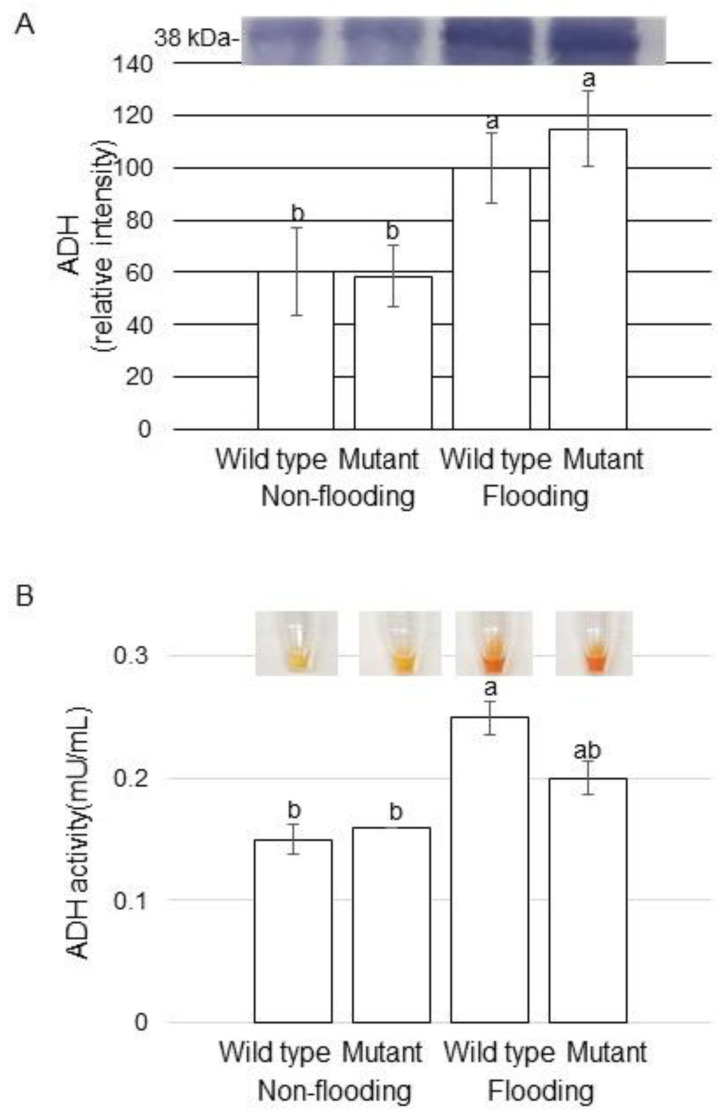
Immunoblot analysis and activity assay of proteins involved in fermentation. (**A**) Proteins were extracted from root including hypocotyl, separated on SDS-polyacrylamide gel by electrophoresis, and transferred onto membrane. The membrane was cross-reacted with anti-ADH antibody. Coomassie brilliant blue staining pattern was used as loading control (Appendix A). The integrated densities of bands were calculated using ImageJ software with 3 independent biological replicates (Appendix A). (**B**) Proteins were extracted from root including hypocotyl and ADH activity assay was performed. Data are shown as means ± SD from 3 independent biological replicates. The different letters indicate significant changes according to one-way ANOVA followed by Tukey’s multiple comparisons (*p* < 0.05).

**Figure 6 ijms-22-09046-f006:**
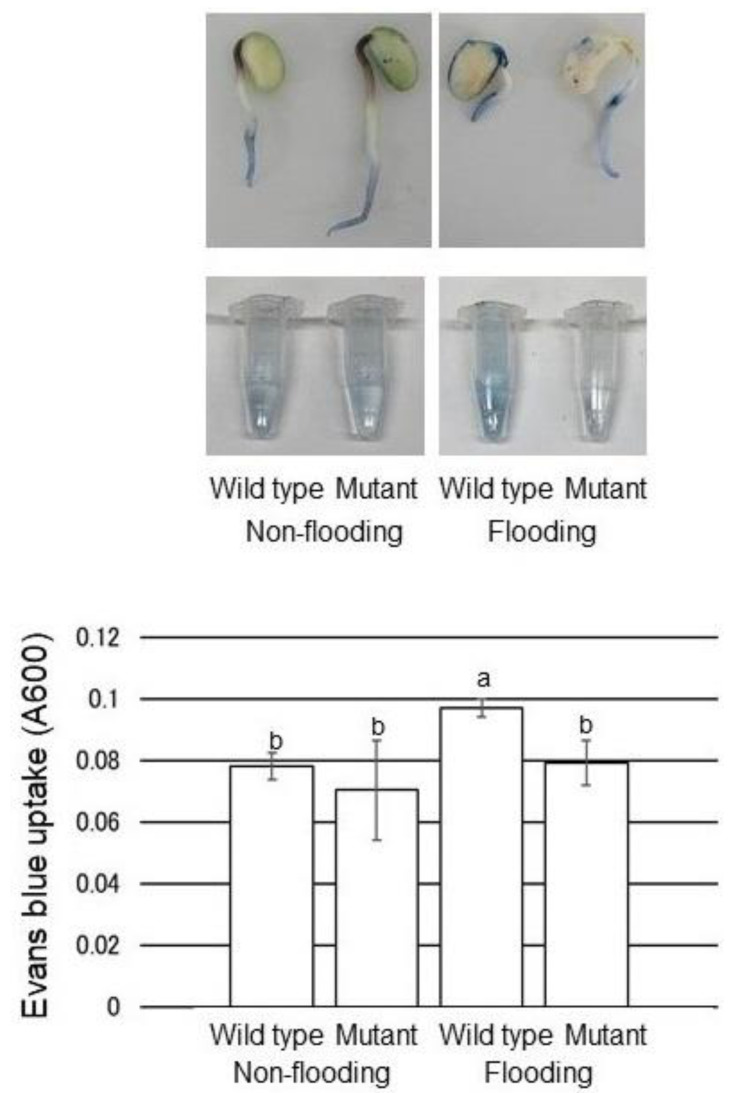
Evaluation of cell death in soybean with Evans-blue staining. After flooding, soybean seedlings were stained with Evans-blue dye. The Evans-blue dye was extracted from the root tip and absorbance was measured at 600 nm. Data are shown as means ± SD from 3 independent biological replicates. The different letters indicate significant changes according to one-way ANOVA followed by Tukey’s multiple comparisons (*p* < 0.05).

## Data Availability

For MS data, RAW data, peak lists and result files have been deposited in the ProteomeXchange Consortium [49] via the jPOST [50] partner repository under data-set identifiers PXD024711.

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
