# Peer review of "Proteomic and Biochemical Analyses of the Mechanism of Tolerance in Mutant Soybean Responding to Flooding Stress"

_ijms, 2021, doi:10.3390/ijms22169046_

Round 1

Reviewer 1 Report

The manuscript „Proteomic and Biochemical Analyses of Tolerant Mechanism in Mutant Soybean Responding to Flooding Stress” presents a very interesting study with new findings that might help improve flooding tolerance in soybean in the future.

Unfortunately, even though the scientific value of the MS is high in my opinion, the writing style and language makes it very confusing in many places.

First of all, the MS must be submitted to English editing service prior to its resubmission. As far as I know IJMS provides such service. Especially, I do not think the term “tolerant mechanism” used by the authors throughout the study is correct. I suggest substituting it with “tolerance mechanism” or “mechanism of tolerance”.

The other problem is the writing style, which is not informative/clear enough in many parts of the MS. It seems as though the writers expect the reader to have certain knowledge about the study before reading the MS, while the reader has none and needs more explanations. Specific remarks:

The abstract needs rewriting, especially the first part – it is very confusing. It is unclear why the mutants were crossed with the parent cultivar and also that the parent cultivar is the same as “wild type” which is also used in the study (e.g. line 91). I think that also the introduction could use a few sentences about the reasons for backcrossing of the mutants to obtain “more useful” soybean. And in the second sentence I cannot tell if the mutant lines are the progeny of this cross, or maybe the mutant lines that were crossed. Also, the phrase “the growth of (plants, mutant lines – depending on the part of the MS, because this phrase was used repeatedly) was improved by the removal of water after flooding” suggests that the growth was better after flooding than before it and I am quite sure it is not what the authors had in mind – or is it? In what way was it improved, compared to what control? If it is simply compared to the wild type, than it should be clearly written, and it is not an improvement. If it is compared to the growth during flooding, than it also should be clearly stated. The same problem is with “recovered” glycoproteins in line 25.

Line 45 – what reports?

Line 48-54 – The authors write about two strategies, introducing properly only one of them. The other is mentioned in the line 54 as if it was introduced before.

Line 58 – “in the case of soybean” is completely unnecessary, the sentence should begin with “although”.

Line 60 – it seems to me that using “whether” or “regardless” would better convey the meaning of this sentence than “when”.

Line 83 – the analysis was performed on Enrei, mutant or the progeny?

Line 96-98 – again “improved” – the authors show only flooding and non-flooding conditions in the figure 2. Where is the improvement? Compared to when?

Line 100 – the length and weight was bigger than in wild type (but not in G4 which the authors seem to forget), but not increased, as those values are smaller than before flooding, and the authors do not show the data for “during the flooding” and “after the flooding” (which could show an increase?), only after the flooding.

Line 105 (Fig 1 description) – what the authors mean by “confirmation”?

Line 122: “As groups” can be omitted.

Line 135: Which proteins?

Line 135 – 144: there are different numbers of differentially changed proteins in each group than in the previous paragraph. It is completely unclear to me why those numbers are different and where they came from.

Paragraph 2.3 – the description of Figure 4 is more clear and informative to me than the description of the results in this paragraph.

Line 165 – again “recovered” , while in figure 4D I can see that the difference between non-flooded and flooded mutant is insignificant (both with “a”), so where is the recovery?

Line 183 (title of the paragraph) – “ADH” should be added in brackets after “alcohol-dehydrogenase, because this is the first time this term is used, and after that the abbreviation is used without explaining.

Line 211 – the loss of root tips is mentioned here casually as if it was described before – but it was not. The sentences before mention “suppression of root growth” and “cell death”, but those do not mean explicitly that the root tip was lost.

Line 212 – by “recovered” do you mean that it regrew after falling off?

Line 221 (title of the paragraph – the mutation has positive effect on plant growth, not the mutant)

Line 224 – the authors write that the analysis of morphological changes of mutant lines were to show the difference of mutant soybean before and after crossing with the wild type, but in the Figure featured (Fig 2), we can see only results for wild type and for mutant – are these mutant progeny from the crossing or mutant “parent”? Either way how is the reader supposed to see the difference between mutant before and after crossing if there is only “mutant” indicated in the Figure.

Line 232 – cross out “before and”

Line 233 – 236: Exactly the same content (maybe in slightly different words) that was already presented in the introduction, such a repetition is completely unnecessary.

Line 237-240: another repetition, this time from the beginning of previous paragraph.

Line 284,285 – flooding with silver nano particles etc.?

Line 314 – mutation might improve, not mutant line

Line 437,438 – as I mentioned before, this suggests that the removal of water after flooding improved the growth of mutant lines (so that it was better than ever before).

Line 442 – again – “recovered” is not the best word to use here

Author Response

The manuscript „Proteomic and Biochemical Analyses of Tolerant Mechanism in Mutant Soybean Responding to Flooding Stress” presents a very interesting study with new findings that might help improve flooding tolerance in soybean in the future.

Unfortunately, even though the scientific value of the MS is high in my opinion, the writing style and language makes it very confusing in many places.

Answer: Thank you very much for your comments and suggestion. Mased on reviewer’s comments and suggestion, this manuscript has been improved with red color in the text.

First of all, the MS must be submitted to English editing service prior to its resubmission. As far as I know IJMS provides such service.

Answer: This article has been corrected by English editing service.

Especially, I do not think the term “tolerant mechanism” used by the authors throughout the study is correct. I suggest substituting it with “tolerance mechanism” or “mechanism of tolerance”.

Answer: Although the word “tolerant mechanism” was suggested native speaker, this word has been changed to “mechanism of tolerance” in all parts of manuscript, now.

The other problem is the writing style, which is not informative/clear enough in many parts of the MS. It seems as though the writers expect the reader to have certain knowledge about the study before reading the MS, while the reader has none and needs more explanations.

Answer: Thank you very much for your comments and suggestion. As suggested, this article has been re-written in all parts of manuscript with red color.

Specific remarks:

The abstract needs rewriting, especially the first part – it is very confusing. It is unclear why the mutants were crossed with the parent cultivar and also that the parent cultivar is the same as “wild type” which is also used in the study (e.g. line 91). I think that also the introduction could use a few sentences about the reasons for backcrossing of the mutants to obtain “more useful” soybean. And in the second sentence I cannot tell if the mutant lines are the progeny of this cross, or maybe the mutant lines that were crossed.

Answer: As suggested, the first part of abstract has been re-rewritten as follows: To investigate the mechanism of flooding tolerance of soybean, flooding-tolerant mutants derived from gamma-ray irradiated soybean were crossed with parent cultivar Enrei for removal of other factors besides the genes related to flooding tolerance in primary generated mutant soybean. 

In the introduction, the detailed explanation has been added as follows: In this study, to remove other factors besides the genes related to flooding tolerance in primary generated mutant soybean, it was crossed with parent cultivar Enrei and flooding-tolerant tests were repeated two times. Using the progeny of this cross, morphological analysis was performed under flooding stress.

In the result section, the first sentence has been corrected as follows: To remove other factors besides the genes related to flooding tolerance in primary generated mutant soybean [16], they were crossed with parent cultivar Enrei.

Also, the phrase “the growth of (plants, mutant lines – depending on the part of the MS, because this phrase was used repeatedly) was improved by the removal of water after flooding” suggests that the growth was better after flooding than before it and I am quite sure it is not what the authors had in mind – or is it? In what way was it improved, compared to what control? If it is simply compared to the wild type, than it should be clearly written, and it is not an improvement. If it is compared to the growth during flooding, than it also should be clearly stated.

Answer: Based on reviewer’s comments, the word “improve” has been changed suitable words in the text with red color. For example, the sentence has been corrected as follows: Although the growth of wild type was significantly suppressed by flooding compared with non-flooding condition, that of mutant lines was better than that of wild type even if it was treated with flooding.

The same problem is with “recovered” glycoproteins in line 25.

Answer: Based on reviewer’s comments, the sentences including the word “recover” has been corrected in the text with red color. For example, the sentence has been corrected as follows: Furthermore, although glycoproteins in wild tyle decreased by flooding compared with non-flooding condition, those in mutant line increased even if it was under flooding stress.

Line 45 – what reports?

Answer: This word has been corrected as “These previous findings”.

Line 48-54 – The authors write about two strategies, introducing properly only one of them. The other is mentioned in the line 54 as if it was introduced before.

Answer: These sentences have been corrected as follows: One of the two adaptive strategies for flooding tolerance is SNORKEL 1/2 dependent escape strategy [6]. It promotes the internode elongation through the stimulation of gibberellin biosynthesis in deep-water rice under flooding stress, and thereby enabling rice grows upward to the water surface for air exchange [6]. Another adaptive strategy is SUBMERGENCE1 dependent quiescence strategy [7]. In a few rice-tolerant varieties against submergence, the elongation of stem and leaf is inhibited by suppressing the increase of ethylene concentration, which decreases the sensitivity of plants to gibberellin [7].

Line 58 – “in the case of soybean” is completely unnecessary, the sentence should begin with “although”.

Answer: As suggested, the sentence has been started with “Although”.  

Line 60 – it seems to me that using “whether” or “regardless” would better convey the meaning of this sentence than “when”.

Answer: After discussion with native speaker, we did not decide to change this sentence. We apologize to this inconvenience.

Line 83 – the analysis was performed on Enrei, mutant or the progeny?

Answer: Thank you very much for your question. This sentence has been changed as follows: In this study, to remove other factors besides the genes related to flooding tolerance in primary generated mutant soybean, it was crossed with parent cultivar Enrei. Using mutant lines crossed with parent cultivar, morphological analysis was performed under flooding stress.

Line 96-98 – again “improved” – the authors show only flooding and non-flooding conditions in the figure 2. Where is the improvement? Compared to when?

Answer: This sentence has been corrected as follows: The length/weight of stem/root of wild type soybean was significantly suppressed by flooding; however, those of 1386-6 (G2) and 1386-9 (G3) lines were better than those of wild type even if they were treated with flooding.

Line 100 – the length and weight was bigger than in wild type (but not in G4 which the authors seem to forget), but not increased, as those values are smaller than before flooding, and the authors do not show the data for “during the flooding” and “after the flooding” (which could show an increase?), only after the flooding.

Answer: We are sorry that the explanation in this paragraph was not appropriate. This paragraph has been corrected with red color.

Line 105 (Fig 1 description) – what the authors mean by “confirmation”?

Answer: We are sorry and this sentence has been corrected as follows: Proteomic results were subsequently confirmed by immuno-blot, enzyme activity, and physiological analyses.

Line 122: “As groups” can be omitted.

Answer: As suggested, the words “As groups” have been omitted.

Line 135: Which proteins?

Answer: We are sorry that this word is not suitable here; so, this word has been deleted.

Line 135 – 144: there are different numbers of differentially changed proteins in each group than in the previous paragraph. It is completely unclear to me why those numbers are different and where they came from.

Answer: This sentence has been corrected with selecting condition of proteins as follows: Furthermore, the abundance of proteins differentially changed with fold change ≥1.5 and ≤1/3 in wild type or mutant line under flooding stress compared with non-flooding were categorized using cellular component of gene-ontology analysis (Figure 3).

Paragraph 2.3 – the description of Figure 4 is more clear and informative to me than the description of the results in this paragraph.

Answer: Thank you very much for your checking. Paragraph 2.2 has been corrected with red color.

Line 165 – again “recovered” , while in figure 4D I can see that the difference between non-flooded and flooded mutant is insignificant (both with “a”), so where is the recovery?

Answer: Based on results from three independent biological replicates, data was analyzed according to one-way ANOVA followed by Tukey’s multiple comparisons. Although averages had difference, significance did not deferent between non-flooded and flooded mutants. As suggested, because the word “recovered” is not so good word, these two sentences have been corrected with red color.

Line 183 (title of the paragraph) – “ADH” should be added in brackets after “alcohol-dehydrogenase, because this is the first time this term is used, and after that the abbreviation is used without explaining.

Answer: As suggested, the word “ADH” has been used after abbreviated, even it is positioned the top of sentence.

Line 211 – the loss of root tips is mentioned here casually as if it was described before – but it was not. The sentences before mention “suppression of root growth” and “cell death”, but those do not mean explicitly that the root tip was lost.

Answer: We are sorry that this explanation was not right. The first sentence has been corrected as follows: To better understand suppression of root growth, the cell death of root-tip was evaluated by Evans-blue staining.

Line 212 – by “recovered” do you mean that it regrew after falling off?

Answer: We are sorry that this explanation was not right. The sentence of this result has been corrected as follows: These results indicated that the suppression of root growth by flooding was caused by cell death of root tip induced by flooding stress in wild type and it has not happened in mutant line.

Line 221 (title of the paragraph – the mutation has positive effect on plant growth, not the mutant)

Answer: Thank you very much for your correction. It has been corrected in the title of this paragraph.

Line 224 – the authors write that the analysis of morphological changes of mutant lines were to show the difference of mutant soybean before and after crossing with the wild type, but in the Figure featured (Fig 2), we can see only results for wild type and for mutant – are these mutant progeny from the crossing or mutant “parent”? Either way how is the reader supposed to see the difference between mutant before and after crossing if there is only “mutant” indicated in the Figure.

Answer: This comparison is between previous report [16], which is “before being crossed with parent cultivar Enrei” and this manuscript, which is “after being crossed with parent cultivar Enrei”. The sentences have been corrected in the paragraph “3.1” with red color.

Line 232 – cross out “before and”

Answer: Thank you very much for your correction.

Line 233 – 236: Exactly the same content (maybe in slightly different words) that was already presented in the introduction, such a repetition is completely unnecessary.

Answer: As suggested, this paragraph has been deleted.

Line 237-240: another repetition, this time from the beginning of previous paragraph.

Answer: As suggested, this paragraph has been deleted.

Line 284,285 – flooding with silver nano particles etc.?

Answer: This example has been re-written as follows: under flooding with silver nano particles with other chemicals, which improved soybean growth

Line 314 – mutation might improve, not mutant line

Answer: Thank you very much for your correction. It has been corrected.

Line 437,438 – as I mentioned before, this suggests that the removal of water after flooding improved the growth of mutant lines (so that it was better than ever before).

Answer: We are sorry that it has been corrected with red color.

Line 442 – again – “recovered” is not the best word to use here

Answer: We are sorry that it has been corrected with red color.

Reviewer 2 Report

This work is a continuation of a previous study by the same team (reference 16 in the manuscript; [16] Komatsu S., Nanjo Y., Nishimura M. 2013. Proteomic analysis of the flooding tolerance mechanism in mutant soybean. J. Proteomics  79, 231-250), in which the authors generated flooding-tolerant mutants in soybean and used a proteomic approach to characterize such mutants. The mutants used in the present analysis are the same as in the previous study (manuscript reference 16). The novelty between the two studies is that in the present work, to eliminate factors other than genes related to flood tolerance in the mutant soybean lines, the selected mutant was crossed with the parent cultivar Enrei, and morphological analysis was performed in the progeny under flooding stress.  Based on the morphological results, a gel-free and label-free proteomic analysis was performed to explore the mechanism of tolerance of the positive growth effects of the flooding-treated mutant soybean. The proteomic results were then confirmed by immunoblot, enzyme activity, and physiological analyses. An analysis of the functions of differentially accumulated proteins in wild type and mutant soybean during flooding stress suggests that regulation of cell death by the fermentation system and glycoprotein folding may be an important factor in the acquisition of flooding tolerance in mutant soybean.
The team involved in this work is internationally recognized for its work in plant proteomics, especially in soybean under environmental stress. The present work is of very high quality both concerning the proteomic analysis and the agronomic implications of the results presented. However, I have a few comments.

1/ Quantitative expression of proteomic results
1.1/ Difference: Does this word refers to a difference in protein accumulation or to a ratio in protein accumulation?
1.2/ In Supplemental Table 1, why is this value becoming negative at the line corresponding to the Transmembrane 9 superfamily member (p. 10)?      
1.3/ Same question as for 1.2 for Supplemental Table 2 (line C6SYK2 Uncharacterized protein, p. 25).   
1.4/ If the numbers listed in the « Difference » column correspond to accumulation ratios between WT or mutant proteins subjected or not to a flooding stress, then indicate in the text at which accumulation ratio values proteins have been considered to be differentially accumulated. Then to facilitate the reading of Supplemental Table 1 and Table 2 please highlight the lines corresponding to proteins not considered to be differentially accumulated, for example by using a grey background.     
1.5/ In both Supplemental Tables 1 and 2, please add a column to list p values. 

2/ Morphological changes. In Figure 2 (l. 107), please show some representative pictures of WT and mutant lines. 

3/ Proteomics 
3.1/ As indicated in point 1 above, it is not clear how the 986 proteins from the WT (l. 125) were selected out of the 7889 proteins presently identified (l. 121). Same question for the 833 proteins from the mutant (l. 128) out of the 6967 proteins presently identified (l. 127). 
3.2/ It would be worth to list the protein changes between WT and mutant under non-flooding or flooding conditions.

4/ Comparison with other studies. There are several proteomic studies on response to flooding stress in soybean, including several studies from the Authors (see below). The results from these studies should be compared to those obtained in the current work. Similarities and differences should be discussed. This could help to draw a unifying picture of the proteome changes during flooding in soybean.

Shi et al. 2008. Cytosolic ascorbate peroxidase 2 (cAPX 2) is involved in the soybean response to flooding. PHYTOCHEMISTRY 69, 1295-1303

Komatsu et al. 2009. A comprehensive analysis of the soybean genes and proteins expressed under flooding stress using transcriptome and proteome techniques. JOURNAL OF PROTEOME RESEARCH 8, 4766-4778

Hashiguchi et al. 2009. Proteome analysis of early-stage soybean seedlings under flooding stress. JOURNAL OF PROTEOME RESEARCH 8, 2058-2069

Alam et al. 2010. Proteome analysis of soybean roots under waterlogging stress at an early vegetative stage. JOURNAL OF BIOSCIENCES 35, 49-62

Nanjo et al. 2010. Comparative proteomic analysis of early-stage soybean seedlings responses to flooding by using gel and gel-free techniques. JOURNAL OF PROTEOME RESEARCH 9, 3989-4002

Alam et al. 2011. Comparative proteomic approach to identify proteins involved in flooding combined with salinity stress in soybean. PLANT AND SOIL 346, 45-62

Komatsu et al. 2011. Comprehensive analysis of mitochondria in roots and hypocotyls of soybean under flooding stress using proteomics and metabolomics techniques. JOURNAL OF PROTEOME RESEARCH 10, 3993-4004

Khatoon et al. 2012. Organ-specific proteomics analysis for identification of response mechanism in soybean seedlings under flooding stress. JOURNAL OF PROTEOMICS 75, 5706-5723

Khatoon et al. 2012. Proteomics analysis of flooding-stressed plant. CURRENT PROTEOMICS 9, 217-231

Komatsu et al. 2012. Comprehensive analysis of endoplasmic reticulum-enriched fraction in root tips of soybean under flooding stress using proteomics techniques. JOURNAL OF PROTEOMICS 77, 531-560

Nanjo et al. 2012. Mass spectrometry-based analysis of proteomic changes in the root tips of flooded soybean seedlings. JOURNAL OF PROTEOME RESEARCH 11, 372-385

Salavati et al. 2012. Analysis of proteomic changes in roots of soybean seedlings during recovery after flooding. JOURNAL OF PROTEOMICS 75, 878-893

Komatsu et al. 2013. Proteomic analysis of the flooding tolerance mechanism in mutant soybean. JOURNAL OF PROTEOMICS 79, 231-250

Komatsu et al. 2013. Label-free quantitative proteomic analysis of abscisic acid effect in early-stage soybean under flooding. JOURNAL OF PROTEOME RESEARCH 12, 4769-4784

Komatsu et al. 2013. Proteomic and biochemical analyses of the cotyledon and root of flooding-stressed soybean plants. PLOS ONE 8, e65301

Komatsu et al. 2013. Proteomic and biochemical analyses of the cotyledon and root of flooding-stressed soybean plants. PLOS ONE 8, e65301

Nanjo et al. 2013. Identification of indicator proteins associated with flooding injury in soybean seedlings using label-free quantitative proteomics. JOURNAL OF PROTEOME RESEARCH 12, 4785-4798

Khan et al. 2014. Quantitative proteomics reveals that peroxidases play key roles in post-flooding recovery in soybean roots. JOURNAL OF PROTEOME RESEARCH 13, 5812-5828

Komatsu et al. 2014. Proteomic and metabolomic analyses of soybean root tips under flooding stress. PROTEIN AND PEPTIDE LETTERS 21, 865-884

Nanjo et al. 2014. Analyses of flooding tolerance of soybean varieties at emergence and varietal differences in their proteomes. PHYTOCHEMISTRY 106, 25-36

Oh et al. 2014. Gel-free proteomic analysis of soybean root proteins affected by calcium under flooding stress. FRONTIERS IN PLANT SCIENCE 5, 559

Oh et al. 2014. Identification of nuclear proteins in soybean under flooding stress using proteomic technique. PROTEIN AND PEPTIDE LETTERS 21, 458-467

Yin et al. 2014. Phosphoproteomics reveals the effect of ethylene in soybean root under flooding stress. JOURNAL OF PROTEOME RESEARCH 13, 5618-5634

Yin et al. 2014. Analysis of initial changes in the proteins of soybean root tip under flooding stress using gel-free and gel-based proteomic techniques. JOURNAL OF PROTEOMICS 106, 1-16

Khan et al. 2015. Proteomic analysis of soybean hypocotyl during recovery after flooding stress. JOURNAL OF PROTEOMICS 121, 15-27

Komatsu et al. 2015. Proteomic techniques and management of flooding tolerance in soybean. JOURNAL OF PROTEOME RESEARCH 14, 3768-3778

Oh et al. 2015. Characterization of proteins in soybean roots under flooding and drought stresses. JOURNAL OF PROTEOMICS 114, 161-181

Nishirnura et al. 2016. Quantitative Proteomics Reveals the Flooding-Tolerance Mechanism in Mutant and Abscisic Acid-Treated Soybean. JOURNAL OF PROTEOME RESEARCH 15, 2008-2025

Wang et al. 2016. Gel-free/label-free proteomic analysis of root tip of soybean over time under flooding and drought stresses. JOURNAL OF PROTEOMICS 130, 42-55

Wang et al. 2016. Gel-free/label-free proteomic analysis of endoplasmic reticulum proteins in soybean root tips under flooding and drought stresses. JOURNAL OF PROTEOME RESEARCH 15, 2211- 2227

Wang et al. 2016. Characterization of S-adenosylmethionine synthetases in soybean under flooding and drought stresses. BIOLOGIA PLANTARUM 60, 269-278

Yin et al. 2016. Nuclear Proteomics Reveals the Role of Protein Synthesis and Chromatin Structure in Root Tip of Soybean during the Initial Stage of Flooding Stress. JOURNAL OF PROTEOME RESEARCH 15, 2283-2298

Oskuei et al. 2017. Proteomic analysis of soybean seedling leaf under waterlogging stress in a time-dependent manner. BIOCHIMICA ET BIOPHYSICA ACTA-PROTEINS AND PROTEOMICS, 1865, 1167-1177

Wang et al. 2017. Organ-specific proteomics of soybean seedlings under flooding and drought stresses. JOURNAL OF PROTEOMICS 162, 62-72

Wang al. 2017. Metabolic profiles of flooding-tolerant mechanism in early-stage soybean responding to initial stress. PLANT MOLECULAR BIOLOGY 94, 669-685

Hashiguchi et al. 2018. Early changes in S-nitrosoproteome in soybean seedlings under flooding stress. PLANT MOLECULAR BIOLOGY REPORTER 36, 822-831

Wang et al. 2018. An integrated approach of proteomics and computational genetic modification effectiveness analysis to uncover the mechanisms of flood tolerance in soybeans. INTERNATIONAL JOURNAL OF MOLECULAR SCIENCES 19, 1301

Wang et al. 2018. Proteomic approaches to uncover the flooding and drought stress response mechanisms in soybean. JOURNAL OF PROTEOMICS 172, 201-215

Lin et al. 2019. Identification of genes/proteins related to submergence tolerance by transcriptome and proteome analyses in soybean. SCIENTIFIC REPORTS 9, 14688

Wang et al. 2020. Proteomic techniques for the development of flood-tolerant soybean. INTERNATIONAL JOURNAL OF MOLECULAR SCIENCES 21, 7497

Wang et al. 2021. Proteomic analysis reveals the effects of melatonin on soybean root tips under flooding stress. JOURNAL OF PROTEOMICS 232, 10406

Author Response

Reviewer 2

This work is a continuation of a previous study by the same team (reference 16 in the manuscript; [16] Komatsu S., Nanjo Y., Nishimura M. 2013. Proteomic analysis of the flooding tolerance mechanism in mutant soybean. J. Proteomics  79, 231-250), in which the authors generated flooding-tolerant mutants in soybean and used a proteomic approach to characterize such mutants. The mutants used in the present analysis are the same as in the previous study (manuscript reference 16). The novelty between the two studies is that in the present work, to eliminate factors other than genes related to flood tolerance in the mutant soybean lines, the selected mutant was crossed with the parent cultivar Enrei, and morphological analysis was performed in the progeny under flooding stress.  Based on the morphological results, a gel-free and label-free proteomic analysis was performed to explore the mechanism of tolerance of the positive growth effects of the flooding-treated mutant soybean. The proteomic results were then confirmed by immunoblot, enzyme activity, and physiological analyses. An analysis of the functions of differentially accumulated proteins in wild type and mutant soybean during flooding stress suggests that regulation of cell death by the fermentation system and glycoprotein folding may be an important factor in the acquisition of flooding tolerance in mutant soybean.
The team involved in this work is internationally recognized for its work in plant proteomics, especially in soybean under environmental stress. The present work is of very high quality both concerning the proteomic analysis and the agronomic implications of the results presented. However, I have a few comments.

Answer: Thank you very much for your checking. Based on reviewers’ comments, this article has been corrected.

1/ Quantitative expression of proteomic results
1.1/ Difference: Does this word refers to a difference in protein accumulation or to a ratio in protein accumulation?
Answer: Thank you very much for your comments. Because “difference” means a fold change in protein abundance, this word has been corrected in Supplemental Tables.

1.2/ In Supplemental Table 1, why is this value becoming negative at the line corresponding to the Transmembrane 9 superfamily member (p. 10)?      
Answer: Thank you very much for your question. It means decreased proteins in wild type soybean under flooding compared with non-flooding.

1.3/ Same question as for 1.2 for Supplemental Table 2 (line C6SYK2 Uncharacterized protein, p. 25).   
Answer: Thank you very much for your question. It means increased proteins in mutant soybean under flooding compared with non-flooding.

1.4/ If the numbers listed in the « Difference » column correspond to accumulation ratios between WT or mutant proteins subjected or not to a flooding stress, then indicate in the text at which accumulation ratio values proteins have been considered to be differentially accumulated. Then to facilitate the reading of Supplemental Table 1 and Table 2 please highlight the lines corresponding to proteins not considered to be differentially accumulated, for example by using a grey background.     
Answer: In the result section “2.2”, the following sentence has been added with red color. As suggested, to facilitate the reading of Supplemental Table 1 and Table 2, grey background has been used to proteins not considered to be differentially accumulated.

1.5/ In both Supplemental Tables 1 and 2, please add a column to list p values. 

Answer: As suggested, a column to list p values has been added in both Supplemental Tables 1 and 2.

2/ Morphological changes. In Figure 2 (l. 107), please show some representative pictures of WT and mutant lines.

Answer: As suggested, picture has been added in Figure 1.

3/ Proteomics 
3.1/ As indicated in point 1 above, it is not clear how the 986 proteins from the WT (l. 125) were selected out of the 7889 proteins presently identified (l. 121). Same question for the 833 proteins from the mutant (l. 128) out of the 6967 proteins presently identified (l. 127). 
Answer: We are sorry that they were not so clear. The criteria for significantly changed proteins has been added as follows: The criteria for significantly changed proteins were 2 or more than 2 matched peptides with a p-value less than 0.05.

3.2/ It would be worth to list the protein changes between WT and mutant under non-flooding or flooding conditions.

Answer: As suggested, lists of the protein changes between WT and mutant under non-flooding or flooding conditions have been prepared as new Supplemental Tables 3 and 4.

4/ Comparison with other studies. There are several proteomic studies on response to flooding stress in soybean, including several studies from the Authors (see below). The results from these studies should be compared to those obtained in the current work. Similarities and differences should be discussed. This could help to draw a unifying picture of the proteome changes during flooding in soybean.

  1. Shi et al. 2008. Cytosolic ascorbate peroxidase 2 (cAPX 2) is involved in the soybean response to flooding. PHYTOCHEMISTRY 69, 1295-1303
  2. Komatsu et al. 2009. A comprehensive analysis of the soybean genes and proteins expressed under flooding stress using transcriptome and proteome techniques. JOURNAL OF PROTEOME RESEARCH 8, 4766-4778
  3. Hashiguchi et al. 2009. Proteome analysis of early-stage soybean seedlings under flooding stress. JOURNAL OF PROTEOME RESEARCH 8, 2058-2069
  4. Alam et al. 2010. Proteome analysis of soybean roots under waterlogging stress at an early vegetative stage. JOURNAL OF BIOSCIENCES 35, 49-62
  5. Nanjo et al. 2010. Comparative proteomic analysis of early-stage soybean seedlings responses to flooding by using gel and gel-free techniques. JOURNAL OF PROTEOME RESEARCH 9, 3989-4002
  6. (Alam et al. 2011. Comparative proteomic approach to identify proteins involved in flooding combined with salinity stress in soybean. PLANT AND SOIL 346, 45-62)
  7. Komatsu et al. 2011. Comprehensive analysis of mitochondria in roots and hypocotyls of soybean under flooding stress using proteomics and metabolomics techniques. JOURNAL OF PROTEOME RESEARCH 10, 3993-4004
  8. Khatoon et al. 2012. Organ-specific proteomics analysis for identification of response mechanism in soybean seedlings under flooding stress. JOURNAL OF PROTEOMICS 75, 5706-5723
  9. Khatoon et al. 2012. Proteomics analysis of flooding-stressed plant. CURRENT PROTEOMICS 9, 217-231
  10. Komatsu et al. 2012. Comprehensive analysis of endoplasmic reticulum-enriched fraction in root tips of soybean under flooding stress using proteomics techniques. JOURNAL OF PROTEOMICS 77, 531-560
  11. Nanjo et al. 2012. Mass spectrometry-based analysis of proteomic changes in the root tips of flooded soybean seedlings. JOURNAL OF PROTEOME RESEARCH 11, 372-385
  12. Salavati et al. 2012. Analysis of proteomic changes in roots of soybean seedlings during recovery after flooding. JOURNAL OF PROTEOMICS 75, 878-893
  13. (Komatsu et al. 2013. Proteomic analysis of the flooding tolerance mechanism in mutant soybean. JOURNAL OF PROTEOMICS 79, 231-250)
  14. (Komatsu et al. 2013. Label-free quantitative proteomic analysis of abscisic acid effect in early-stage soybean under flooding. JOURNAL OF PROTEOME RESEARCH 12, 4769-4784)
  15. Komatsu et al. 2013. Proteomic and biochemical analyses of the cotyledon and root of flooding-stressed soybean plants. PLOS ONE 8, e65301
  16. (Komatsu et al. 2013. Proteomic and biochemical analyses of the cotyledon and root of flooding-stressed soybean plants. PLOS ONE 8, e65301)
  17. Nanjo et al. 2013. Identification of indicator proteins associated with flooding injury in soybean seedlings using label-free quantitative proteomics. JOURNAL OF PROTEOME RESEARCH 12, 4785-4798
  18. Khan et al. 2014. Quantitative proteomics reveals that peroxidases play key roles in post-flooding recovery in soybean roots. JOURNAL OF PROTEOME RESEARCH 13, 5812-5828
  19. Komatsu et al. 2014. Proteomic and metabolomic analyses of soybean root tips under flooding stress. PROTEIN AND PEPTIDE LETTERS 21, 865-884
  20. Nanjo et al. 2014. Analyses of flooding tolerance of soybean varieties at emergence and varietal differences in their proteomes. PHYTOCHEMISTRY 106, 25-36
  21. Oh et al. 2014. Gel-free proteomic analysis of soybean root proteins affected by calcium under flooding stress. FRONTIERS IN PLANT SCIENCE 5, 559
  22. Oh et al. 2014. Identification of nuclear proteins in soybean under flooding stress using proteomic technique. PROTEIN AND PEPTIDE LETTERS 21, 458-467
  23. Yin et al. 2014. Phosphoproteomics reveals the effect of ethylene in soybean root under flooding stress. JOURNAL OF PROTEOME RESEARCH 13, 5618-5634
  24. Yin et al. 2014. Analysis of initial changes in the proteins of soybean root tip under flooding stress using gel-free and gel-based proteomic techniques. JOURNAL OF PROTEOMICS 106, 1-16
  25. Khan et al. 2015. Proteomic analysis of soybean hypocotyl during recovery after flooding stress. JOURNAL OF PROTEOMICS 121, 15-27
  26. Komatsu et al. 2015. Proteomic techniques and management of flooding tolerance in soybean. JOURNAL OF PROTEOME RESEARCH 14, 3768-3778
  27. Oh et al. 2015. Characterization of proteins in soybean roots under flooding and drought stresses. JOURNAL OF PROTEOMICS 114, 161-181
  28. Nishirnura et al. 2016. Quantitative Proteomics Reveals the Flooding-Tolerance Mechanism in Mutant and Abscisic Acid-Treated Soybean. JOURNAL OF PROTEOME RESEARCH 15, 2008-2025
  29. Wang et al. 2016. Gel-free/label-free proteomic analysis of root tip of soybean over time under flooding and drought stresses. JOURNAL OF PROTEOMICS 130, 42-55
  30. (Wang et al. 2016. Gel-free/label-free proteomic analysis of endoplasmic reticulum proteins in soybean root tips under flooding and drought stresses. JOURNAL OF PROTEOME RESEARCH 15, 2211- 2227)
  31. Wang et al. 2016. Characterization of S-adenosylmethionine synthetases in soybean under flooding and drought stresses. BIOLOGIA PLANTARUM 60, 269-278
  32. Yin et al. 2016. Nuclear Proteomics Reveals the Role of Protein Synthesis and Chromatin Structure in Root Tip of Soybean during the Initial Stage of Flooding Stress. JOURNAL OF PROTEOME RESEARCH 15, 2283-2298
  33. Oskuei et al. 2017. Proteomic analysis of soybean seedling leaf under waterlogging stress in a time-dependent manner. BIOCHIMICA ET BIOPHYSICA ACTA-PROTEINS AND PROTEOMICS, 1865, 1167-1177
  34. Wang et al. 2017. Organ-specific proteomics of soybean seedlings under flooding and drought stresses. JOURNAL OF PROTEOMICS 162, 62-72
  35. (Wang al. 2017. Metabolic profiles of flooding-tolerant mechanism in early-stage soybean responding to initial stress. PLANT MOLECULAR BIOLOGY 94, 669-685)
  36. Hashiguchi et al. 2018. Early changes in S-nitrosoproteome in soybean seedlings under flooding stress. PLANT MOLECULAR BIOLOGY REPORTER 36, 822-831
  37. Wang et al. 2018. An integrated approach of proteomics and computational genetic modification effectiveness analysis to uncover the mechanisms of flood tolerance in soybeans. INTERNATIONAL JOURNAL OF MOLECULAR SCIENCES 19, 1301
  38. (Wang et al. 2018. Proteomic approaches to uncover the flooding and drought stress response mechanisms in soybean. JOURNAL OF PROTEOMICS 172, 201-215)
  39. Lin et al. 2019. Identification of genes/proteins related to submergence tolerance by transcriptome and proteome analyses in soybean. SCIENTIFIC REPORTS 9, 14688
  40. (Wang et al. 2020. Proteomic techniques for the development of flood-tolerant soybean. INTERNATIONAL JOURNAL OF MOLECULAR SCIENCES 21, 7497)
  41. Wang et al. 2021. Proteomic analysis reveals the effects of melatonin on soybean root tips under flooding stress. JOURNAL OF PROTEOMICS 232, 10406

Answer: Thank you very much for your useful suggestion. In suggested publication for discussion, because no14 and no15 are the same article, we have used 40 publications. In 40 publications, because 10 publications, which are no1 to no10, are flooding response mechanism, they have been introduced in the section of Introduction using review article, which are no38, no40, Komatsu et al 2013a, and Komatsu et al 2015. So, the results from remaining 30 publications or additional publications have been compared to those obtained in the current work with similarities and differences. Additional discussion has been marked in red color.